# A Simple and Fast Method for the Formation and Downstream Processing of Cancer-Cell-Derived 3D Spheroids: An Example Using Nicotine-Treated A549 Lung Cancer 3D Spheres

**DOI:** 10.3390/mps6050094

**Published:** 2023-10-04

**Authors:** Irida Papapostolou, Florian Bochen, Christine Peinelt, Maria Constanza Maldifassi

**Affiliations:** Institute of Biochemistry and Molecular Medicine, University of Bern, 3012 Bern, Switzerland; irida.papapostolou@unibe.ch (I.P.); florian.bochen@unibe.ch (F.B.); christine.peinelt@unibe.ch (C.P.)

**Keywords:** 3D cell culture, three-dimensional cell culture, low-attachment surface, cancer cells, spheroids, nicotine, nicotinic acetylcholine receptors

## Abstract

Although 2D in vitro cancer cell cultures have been used for decades as a first line-of-research tool to investigate antitumoral drugs and treatments, their use presents many drawbacks, including the poor resemblance of such cultures to the characteristics of in vivo tumors. To mitigate these drawbacks, 3D culture models have emerged as a more representative alternative. Cancer cells cultured as 3D structures have the advantage of resembling solid tumors in their architecture and in their resistance to chemotherapeutic drugs, in part because of restrained drug penetration. Additionally, these 3D structures create a more physiological environment for the study of immune cell invasion and migration, comparable to solid tumors. In this paper, we describe a fast and cost-effective step-by-step protocol for the generation of 3D spheres using ultra-low-attachment (ULA) multiwell plates, which can be incorporated into the normal workflow of any laboratory. Using this protocol, spheroids of different human cancer cell lines can be obtained and can then be characterized on the basis of their morphology, viability, and expression of specific markers.

## 1. Introduction

Two-dimensional (2D) cell culture systems are widely used for drug discovery and cancer research, but three-dimensional (3D) cell culture systems such as spheroids and organoids are now employed more frequently as in vitro cancer models. A common question that arises when thinking of changing or starting to employ such 3D models is: what advantages do these models have? Firstly, amongst the models used, there is a fine difference between 3D spheroids and 3D organoids. The former is derived from monolayers of epithelial cells, which are cultured in low-attachment conditions that promote the formation of cell aggregates; the latter originates from tissues, or from adult or embryonic stem cells, using specific cell culture conditions [1]. In addition, spheroids may consist entirely of cancer cells (homotypic spheroids), or they can be co-cultured with other cellular types (heterotypic spheroids), allowing the study of more complicated cellular interactions within tumors than is possible with 2D cultures [2]. For example, 3D structures are an ideal physiological environment to study immune cell penetration [3], and in co-cultures using endothelial cells they enable the investigation of tumor vascularization [4].

The benefit of using 3D spheroids as an in vitro cellular model is that they simulate numerous features of in vivo solid tumors [2]. Among these features are (i) different cell layers, with external proliferating cells and an internal necrotic core; (ii) oxygen and nutrient gradients within the structure; (iii) protein and gene expression profiles that resemble those of solid tumors (for example, there is high resemblance in the expression of specific epithelial-to-mesenchymal transition markers (EMTs); (iv) a low internal pH; (v) cell–cell and cell–matrix interactions; and (vi ) drug penetration characteristics similar to those of solid tumors [2,3,4,5]. Some studies have also reported that 2D and 3D cell cultures exhibit different degrees of sensitivity to cancer drugs, so 3D cultures are more appealing for pharmaceutical studies. For example, spheroids have been used to screen putative new anticancer compounds and to assess the response of individuals to chemotherapy; this can be seen as a step closer to precision chemotherapy [1]. A detailed comparison of 2D and 3D cultures, in which their properties and benefits are described, can be read in [2,6,7]. Moreover, 3D cellular models are daily gaining more relevance as an alternative to the use of animal models to implement the 3Rs principles of Replace, Refine, and Reduce [8]. All the above features explain why 3D structures, which can partly resemble solid tumors, are now increasingly being used instead of 2D cultures for in vitro tumor models in cancer research [9].

As mentioned above, 3D models can be created using cells derived from tissue samples; they can also be created using induced pluripotent stem cells (iPSCs) or cell lines [10]. Whereas the use of cells derived from patient samples to generate these 3D structures gives rise to a more representative native tumor environment, the use of commercially available cell lines offers the advantages of easy handling and access [8]. The methods used to construct these 3D cultures can basically be divided into the two categories of scaffold-based and scaffold-free procedures [11]. Additionally, the choice of the methodology used to generate the ideal 3D model largely depends upon the specific research question being addressed and the capacity of the laboratory involved. As a general guideline, an in-depth analysis evaluating different 3D culture methods can be found in [2,4,8]. Among the most-used non-scaffold-based protocols are the production of spheroids or cellular aggregates [2]. Commonly used procedures for the generation of these spheres include the use of hanging-drop methods or ultra-low-attachment (ULA) plates, which are of low cost and enable the production of many spheroids with reproducible properties [2].

After a 3D spheroid culture is obtained, a number of tools are available to evaluate and characterize the model. Among the assays used to study spheroid structure/function, low-cost and fast techniques include (i) viability determinations, (ii) image-based analyses, (iii) measurements of protein and mRNA expression, and (iv) mechanical properties. Cell viability assays, which can be a direct or indirect measure of live/dead cells, include dye-exclusion assays such as trypan blue, fluorimetric assays with calcein and propidium iodide, and luminometric determinations [7]. By measuring sphere viability, efficacy data of diverse drugs or treatments can be obtained [7]. Image-based methods may be used not only to determine the size, morphology, architecture, and metabolic status of spheres, but also their viability, if fluorescent probes are used [7]. Quantification of diverse proteins and gene expression levels can be accomplished using routine techniques such as immunohistochemistry, real-time quantitative transcription polymerase chain reaction (qPCR), and Western blotting (WB) [10]. All these methods have been used successfully to study drug resistance mechanisms, for drug screening, and to analyze diverse signaling events and physical properties in spheroids derived from different human cancer cell lines [12,13,14,15,16,17,18]. Spheres can also be used to investigate the tumoricidal activity of immune cells by allowing for the measurement of parameters such as invasion and cytotoxicity in co-culture systems [4]. A detailed overview of various techniques used to assess spheroid structure and function can be found in [2,8,10].

In this paper, we report an optimized, efficient, cheap, and fast method for generating spheres derived from different human cancer cell lines. This method can be easily integrated into a 2D cell culture workflow in any laboratory and has previously been used by our group to generate spheres from the DU145 prostate cancer cell line [19]. The protocol described uses ultra-low-attachment (ULA) multiwell plates with U-bottoms for spheroid generation. These ULA plates decrease the capability of the cells to adhere to the well, as a result encouraging cell–cell interactions, thereby leading to the formation of 3D spheroids. An advantage of using this proposed method is that every spheroid obtained will have the same volume and growth rates and are more reproduceable as compared to other methods where spheroids of different sizes are grown. Although we propose cell-seeding numbers that are appropriate for the majority of cell types, we recommend that seeding numbers are determined experimentally. We also provide optimized protocols to characterize spheroid viability and expression of marker genes by obtaining mRNA and using qPCR. As an example, we show that by using these procedures we can study whether activation of nicotinic acetylcholine receptors (nAChRs) by nicotine exposure can modify the above-mentioned parameters in A549 lung cancer-derived spheres.

## 2. Experimental Design

Spheres were obtained from diverse commercially available cell lines (Table 1) using ultra-low-attachment (ULA) plates. The cell lines shown here are of routine use in our laboratory. Alternative common cell lines that can also form spheres are described in [12]. The basic steps consisted of cell counting, appropriate seeding into ULA plates, and optional centrifugation. Afterwards, these spheres were subjected to diverse structural and functional assays, including viability and protein expression using qPCR, as detailed below. A visual scheme illustrating the protocol used is presented in Figure 1.

## 3. Materials and Methods

### 3.1. Equipment and Instruments

Echo Revolve microscope for imaging (Discover Echo, San Diego, CA, USA);Centrifuge provided with a microliter plate rotor (optional);Cell culture incubator with a humidified environment at 5% CO_2_ and 37 °C;Tecan Spark (Tecan, AG, Männedorf, Switzerland);96-well BIOFLOAT ULA ultra-low-attachment plates (FaCellitate, Mannheim, Germany);Opaque-walled 96-well plates;NanoPhotometer (AxonLab, Baden, Switzerland);ViiA 7 Real-Time PCR System (Applied Biosystems/Thermo Fisher Scientific, Schwerte, Germany).

### 3.2. Reagents

PBS (Gibco, Waltham, MA, USA);FBS (Gibco, Waltham, MA, USA);Trypsin (Gibco, Waltham, MA, USA);Medium: MEM (Gibco, Waltham, MA, USA);Medium: DMEM (Gibco, Waltham, MA, USA);CellTiter-Glo 3D assay (Promega, Madison, WI, USA);RNeasy Plus Micro Kit (Qiagen, Hilden, Germany);Nicotine (Tocris, Bristol, UK);Mecamylamine (MEC) (Tocris, Bristol, UK);High-Capacity cDNA Reverse Transcription Kit (Thermo Fisher Scientific, Dreieich, Germany);TaqMan Gene Expression Assay and Master Mix (Thermo Fisher Scientific, Dreieich, Germany);L-glutamine (Gibco, Waltham, MA, USA);MEM Non-Essential Amino Acids Solution (Gibco, Waltham, MA, USA).

### 3.3. Cell Culture

The following human cancer cell lines were used:MCF7 (breast cancer);A549 (lung cancer);DU145 (prostate cancer);HCT116 (colon cancer);Lovo (colon cancer).

All cell lines were obtained from ATCC and cultivated according to ATCC instructions.

## 4. Detailed Procedure

### 4.1. Stepwise Generation of Spheres

Note: Using this step-by-step protocol, individual spheres will be obtained in each well of a ULA FaCellitate 96-well plate. Corning low-attachment round-bottom plates may also be used as an alternative.

Culture the cells aseptically, and once the flask is at 70–80% confluence, proceed to dissociation. Note: cells with a passage number of below 20 are recommended for spheroid generation.Aspirate corresponding cell medium.Wash the cells with sterile PBS to remove any traces of serum.Add appropriate amounts of pre-warmed trypsin–EDTA solution to cover the cells.Place the flask in a humidified incubator at 5% CO_2_ and 37 °C for 5 min. Inspect the cells using a microscope to ensure complete cell detachment.To stop the action of trypsin, add the corresponding medium with 10% serum.Gently resuspend the cell suspension until no cell clumps are visible.Count the number of cells, e.g., using an automated cell counter.Initially titrate the optimal number of cells needed to form spheres depending on cell type. We recommend starting with seeding at 1000–2500–5000 and 10,000 cells per well. Note: the reader can also refer to [12], in which seeding numbers of 60 diverse cell lines for successful sphere formation using ULA plates are provided.Dilute cells to the desired cell concentration/number per ml. Note: take into consideration that the final volume per well is 200 µL.Next, add 100 µL of pre-warmed medium using a multichannel pipette to each well of a ULA 96-well plate.Then, seed the selected number of cells, adding the corresponding 100 µL of cell suspension into the wells of the ULA plate. Note: these ULA plates can also be used to form individual patient-derived spheroids from tumor tissue [20].In case of long incubation times, we recommend filling the non-used wells of the plate with 1X PBS or sterile water to prevent evaporation of medium during incubation.Centrifuge the plate at 300 rpm for 5 min. Note: This step assists cell aggregation, leading to uniform spheroid formation. Change centrifuge parameters as needed for each cell line.Place the multiwell plate in the incubator at 37 °C and 5% CO_2_. Note: Spheres must be monitored by using a microscope every day. For most cell lines using this protocol, spheres are formed after 48–72 h and are stable for up to two weeks.Change medium twice per week. To disturb the spheres as little as possible, carefully remove 80 µL using a multichannel pipette (optional) and then add 100 µL of fresh full medium. Afterwards, centrifuge the plate at 300 rpm for 5 min and return it to the cell culture incubator to help maintain sphere integrity.Note: If spheres are going to be used for a drug screening assay, molecules can be co-incubated with the spheres from day 1 of seeding (this study, see below). Alternatively, spheres can be grown for 3–7 days to allow final organoid formation before drug testing, as described in [12,20]. Here, one has the possibility to use several assays, including sphere growth, which can be helpful to assess potential anti-cancer compounds [14].Afterwards, spheres can be subjected to viability assays and qRT-PCR analyses, as detailed below. Furthermore, they can be used for Western blotting; for the protocol, please refer to [16].Note: Images of the spheroids should be collected every day using brightfield settings and a 10× magnification on an Echo Revolve microscope, using annotation tools to determine length and area. Alternatively, using an inverted transmission light microscope, one can calculate the corresponding volume of the sphere by using the mathematical formula as in [14].

### 4.2. Downstream Application I: Viability Assay of Spheres

Note: In this protocol, ATP content, which is proportional to the number of live cells, is used as a measure of cell viability. Here, the CellTiter-Glo 3D (Promega) reagent is used. It is optimized for 3D-specific analysis of viability.

Prepare spheres as described above. Note: spheroid size and the number of days in culture must be determined experimentally for each cell line.Thaw the CellTiter-Glo 3D reagent at 4 °C overnight.Place the reagent at room temperature (RT) 30 min prior to use. Mix before using.Pick/transfer each individual spheroid together with 100 µL of medium from the 96-well ULA plate to an individual well of an opaque-walled 96-well plate. Note: as the spheres must not be disturbed in this process, use 1 mL pipette tips (cut off the front part of each tip to create a larger opening) for handling.Add 100 µL of CellTiter-Glo 3D reagent.Mix by vigorous shaking for 5 min to induce sphere/cell lysis. Note: This step ensures the optimal readout from all cells composing the sphere structure. As this assay relies on the lysis of the sphere, it is considered as an endpoint measurement [4].Incubate the plate at RT for 25 min.Record luminescence according to the manufacturer’s instructions.Note: A minimum of three technical replicates per condition must be used. In addition, a blank control for each respective culture medium of readout must be included.Note: plate shaking, incubation, and luminescence recording may be performed using a Tecan Spark.Note: Optionally, the concentration of ATP from the spheres may be determined. In this case, an ATP standard curve must be created.

### 4.3. Downstream Application II: Obtaining RNA from Spheres

Note: This protocol is for extracting RNA from spheres using the RNeasy Plus Micro Kit (Qiagen, Germany). RNA extraction from spheres using Trizol reagent may also be used as an alternative [14,16].

Place 15 spheres per condition/treatment into an individual Eppendorf tube. Note: do not disturb the spheres in this process, use 1 mL pipette tips with cut tips for handling.Centrifuge the tube at 800 rpm for 10 min.Carefully remove the supernatant without disturbing the pellet.Wash the pellet with 1× PBS.Carefully remove the supernatant without disturbing the pellet. Note: at this point, the sphere pellet obtained can be stored at −80 °C for later use if necessary.Continue with sphere RNA extraction using the RNeasy Plus Micro Kit as per the manufacturer’s instructions. In brief:Into each tube, add 350 µL of RLT lysis buffer provided with the kit.Homogenize by pipetting, transfer to the gDNA Eliminator spin column, and centrifuge for 30 s at top speed.Add an equal volume of 70% ethanol to the above obtained flow-through, place in a new RNeasy column, and centrifuge for 30 s at top speed.To the spin column, add 700 µL RW1 buffer, and centrifuge for 30 s at top speed.Next, add 500 µL RPE buffer, and centrifuge for 30 s at top speed.Add 500 µL 80% ethanol to the column, then centrifuge for 2 min at top speed.Place the column in a new tube and elute the purified RNA using 10 µL RNAse-free water.Measure concentration and purity using a nanophotometer. Note: the size of the spheres may affect the yield of extracted RNA.Store the samples at −80 °C until later use. RNA obtained can be used for the analysis of the expression of diverse genes using qPCR.Note: the number of spheres to be harvested per condition must be optimized for each cell line.

## 5. Expected Results

### 5.1. Generation of Spheres

In our laboratory, using the detailed protocol set out above, diverse cell lines have been proven able to develop into spheres (Table 1). Figure 2 shows representative 2D culture and 3D spheroid images obtained from the Lovo and HCT116 human cancer cell lines using a ULA 96-well plate from FaCellitate. Spheroids were imaged at different time points using an Echo Revolve microscope.

### 5.2. Viability Assay of Spheres: Example from Nicotine-Treated A549-Derived Spheres

We next used our protocol to investigate if nicotine affects the viability of the lung cancer cell line A549 and the prostate cancer cell line DU145 when cultured in 3D. Previous studies using 2D culture systems have shown that nicotine is able to activate endogenously expressed nicotinic acetylcholine receptors (nAChRs) present in A549 cells, which, as a result, increased their proliferation and viability and showed resistance to apoptosis induced by chemotherapeutic drugs [21,22]. In this example, A549- and DU145-derived spheres were cultured in the presence of 0.1 to 10 µM of nicotine from day 1 of 3D culture, as in step 17 of Section 4.1 above. After 7 days of treatment, the sphere area was measured and viability was determined (Figure 3A–D). A representative A549-derived sphere is shown in Figure 3E.

As can be seen in Figure 3A,B, sphere area was not affected by nicotine exposure in either A549- or DU145-derived spheres. Next, we used the CellTiter-Glo 3D assay to measure the levels of ATP, as this directly correlates with viability. As can be seen in Figure 3C, A549 spheroid viability was increased with nicotine, but only when organoids were exposed to 10 µM nicotine. When spheres were incubated with 10 µM nicotine in the presence of the non-specific nAChR inhibitor mecamylamine (MEC) at a concentration of 10 µM, viability was reduced. This showed that the effect of nicotine on viability was achieved through activation of nAChRs. On the contrary, in DU145 spheroids treated with nicotine, no change in viability was recorded (Figure 3D).

### 5.3. Obtaining RNA from Spheres: Example Determination of CD47 Expression from Nicotine-Treated Spheres

Subsequently, we sought to analyze whether nicotine could induce CD47 expression in 3D cultures of A549 cells. CD47 is an immune suppressor protein that is known to be up-regulated in lung cancer cells exposed to nicotine [23]. Thus, RNA from nicotine-treated spheres was obtained as detailed above. Then, reverse transcription of 0.5 µg RNA was performed, cDNA was diluted, and gene expression was evaluated using the TaqMan Gene Expression Assay (qPCR). The following program was used: 2 min activation at 50 °C; 10 min hold at 95 °C; 40 cycles of 15 s denaturation at 95 °C; and 1 min annealing at 60 °C. The results were analyzed using the ΔΔCt method, and expression levels were normalized to the housekeeping gene TATA-binding protein. The following Thermo Fisher primer-probes were used to quantify expression levels (Table 2).

As can be observed in Figure 3F, in the A549 2D culture system (left), 10 µM nicotine induced the expression of CD47. This effect was reversed when cells were incubated with 10 µM nicotine in the presence of the non-specific nAChR inhibitor mecamylamine (MEC) at a concentration of 10 µM. When compared to 3D spheres (Figure 3F, right), nicotine also induced up-regulation of CD47, which was reversed in the presence of MEC. These results demonstrate that the up-regulation of this protein was caused by nAChR signaling in 2D and 3D cultures of this cell line. On the contrary, for the DU145 cell line, no difference in CD47 expression was obtained when cells were incubated with 10 µM nicotine in 2D or 3D formats.

Troubleshooting Tips

It is essential to use a low-passage cell line when making cell-derived spheres, and always limit the number of passaging of cell lines.As mentioned above, e.g., Corning ultra-low-attachment round-bottom plates can be used as an alternative to FaCellitate ULA 96-well plates.Although we recommend using centrifugation steps, it is also possible to change parameters of time and rpm. If no spheroid growth is observed, it may be necessary to use less intense centrifugation conditions.Spheres can be stained using propidium iodine PI and annexin labeling; this can give a good readout of tumorsphere health. Additionally, it can be used to monitor the results of drug/compound screening.Our protocol has a limitation, as one sphere is formed per well. To obtain a high number of spheres for large-scale experiments, such as protein/RNA extraction, it requires the multiple seeding of numerous wells in parallel. The number of such wells must be determined experimentally.In Table 1, optimal numbers of cells were determined experimentally in our laboratory, where a lower or higher number yielded in 3D structures with reduced viability (the structure disintegrated after 2–7 days of culture) was determined visually using an appropriate microscope.

## 6. Conclusions

The protocol we describe for the culture of cells in a 3D format sphere is easy, not time consuming, and affordable to every laboratory. We also show that following this protocol, a series of various applications may be performed to analyze sphere characteristics. 

## Figures and Tables

**Figure 1 mps-06-00094-f001:**
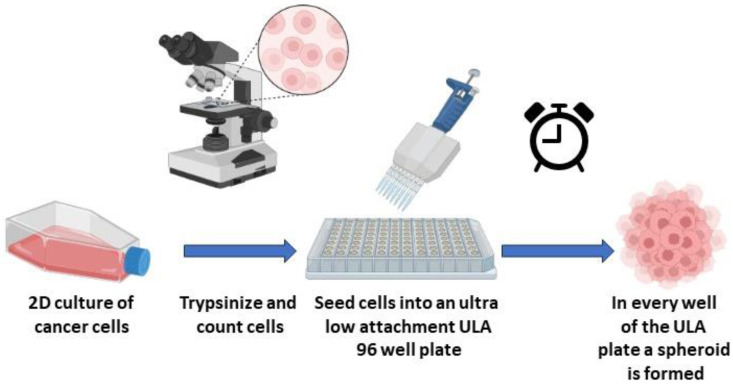
Summary of spheroid generation workflow using a simple, fast, and non-expensive protocol. Cell lines usually cultivated in 2D format can be used to generate 3D structures referred to as spheres (tumorspheres). For the protocol in the current study, we used special ultra-low-attachment (ULA) plates together with specific centrifugation steps. Using this method, we show here the obtention of spheres from 5 different human cancer cells and diverse applications. The image was created with BioRender.com (accessed on 1 February 2023).

**Figure 2 mps-06-00094-f002:**
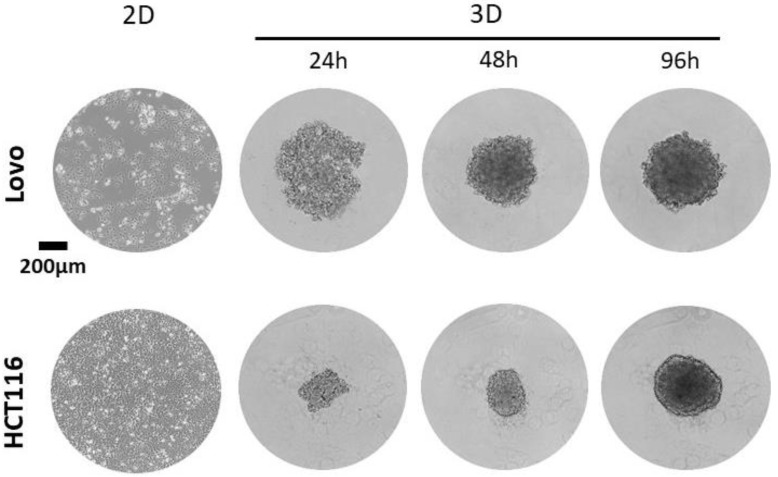
Representative 2D and 3D spheroid images from the human colon cancer cell lines Lovo (**top**) and HCT116 (**bottom**). Cells were seeded in ULA 96-well plates at a density of 4000 (Lovo) or 1000 (HCT116) cells per well, and brightfield images were obtained after 24, 48, and 96 h using a 4× objective. After 48 h following seeding, the formation of sphere structures was evident for both cell types shown. Scale bar 200 µm.

**Figure 3 mps-06-00094-f003:**
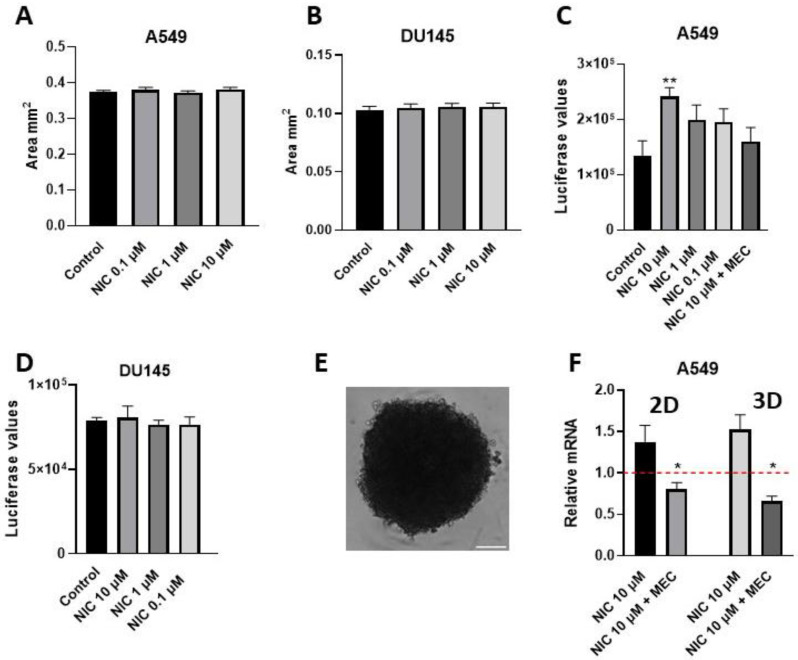
Three-dimensional spheroids: nicotine’s effect upon sensitivity and up-regulation of the immune-suppressive protein CD47. Area in mm^2^ of A549 (**A**) and DU145 (**B**) 3D spheroids after 7 days of growth with exposure to nicotine concentrations of 0.1–10 µM. The bar diagram shows the mean ± SEM. Effect of nicotine exposure upon viability after 7 days of treatment in A549- (**C**) and DU145- (**D**) derived spheroids. The bar diagram shows the mean ± SEM. Nicotine (10 µM) increased viability of A549 3D spheroids; this was reversed when they were co-cultured in the presence of MEC (10 µM). (**E**) Representative brightfield A549 3D spheroid image obtained 7 days after seeding using an ECHO microscope: 10× objective. Scale 200 µm. (**F**) Effects of nicotine on the expression of the immunosuppressive protein CD47 in A549 2D (**left**) vs. 3D (**right**) culture models. The gene expression profile was investigated using qPCR analysis. Bar graphs show the relative expression ± SEM normalized to the corresponding expression in non-treated 2D or 3D cultures (NT: set to a value of 1, red dashed line). Measurements were made using 3–5 independent experiments performed in triplicate. Cells were incubated with nicotine (10 μM) for 7 days, with or without the inhibitor MEC (10 µM). Statistical significance was analyzed using the Kruskal–Wallis test for non-parametric data using GraphPad Prism (GraphPad 9.1.1 Software) software. * *p*-value of <0.05; ** *p*-value of <0.01.

**Table 1 mps-06-00094-t001:** Cell lines and optimal cell numbers used to create spheres within this study.

Cell Line	Origin	Number of Cells Seeded Per Well ^a^
MCF7	Human, breast cancer	5000
A549	Human, lung cancer	10,000
DU145	Human, prostate cancer	10,000
HCT116	Human, colon cancer	1000
Lovo	Human, colon cancer	4000

^a^ Optimal number of cells used per well to generate spheres following this protocol was experimentally determined in our laboratory. In all cases, sphere formation was observed 48–72 h after seeding.

**Table 2 mps-06-00094-t002:** Corresponding proteins and primer-probes used in the present study.

Protein	Gene Name	Probe ID
CD47	CD47	Hs00179953_m1
TATA-binding protein	TBP	Hs00427621_m1

## Data Availability

The data presented in this study are available on request from the corresponding author.

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
