# Peer review of "A Simple and Fast Method for the Formation and Downstream Processing of Cancer-Cell-Derived 3D Spheroids: An Example Using Nicotine-Treated A549 Lung Cancer 3D Spheres"

_mps, 2023, doi:10.3390/mps6050094_

Round 1

Reviewer 1 Report

In this manuscript, Papapostolou et al, describe a simple and fast method for the formation and downstream processing of cancer-cell-derived 3D spheroids using ultra-low adherence multiwell plates. Moreover, Authors demonstrate that spheroids are useful to analysis several cellular aspect induced by stimulation with specific ligands, such as nicotine.

The protocol is well written and easy to understand. Therefore, is suggest to be accepted.

Reviewer 2 Report

The manuscript authored by Irida et al. presents a valuable investigation into the utilization of a 3D culture system employing U shape ultra low attachment (ULA) plates. This method is employed to assess cell viability and marker gene expression within spheroids, particularly in the context of evaluating the influence of nicotine exposure on A549 lung cancer-derived spheres.

Overall, the manuscript provides a well-structured protocol commonly utilized in laboratory settings. However, there are certain areas that require refinement and additional details for improvement.

The following specific comments and suggestions are offered to enhance the paper:

1.      Line 124: Please provide insights into the rationale behind the selection of the specific cell lines listed in the table. In my experience, only a few cell lines have the capability to form spheroids, and some may not yield the desired results. If certain cell lines do not work, it would be beneficial if you could suggest alternative approaches or recommendations to proceed with the experiments.

2.      Line 202: Clarify the primary objective of the centrifugation step mentioned in this section. Understanding the purpose of this step is essential for the reader to grasp its significance within the protocol.

3.      Line 205: Consider discussing the potential impact of drug treatments on spheroid formation. Evaluating how various drugs may affect the formation process can provide valuable insights into experimental outcomes.

4.      Line 208: It would be advantageous to mention the inclusion of Western blot analysis (WB) in addition to other assays.

5.      Line 212: Could the authors provide any mathematical formulas or techniques for measuring spheroid size? This would facilitate a standardized and objective assessment of spheroid dimensions.

6.      Section 5.2: Including images of spheroids under different experimental conditions and providing a detailed protocol for Nicotine treatment would greatly aid in understanding the methodology. The visual representation can be instrumental in conveying key findings.

7.      Figure 3, Panel F: If applicable, please compare the results obtained in A549 with those from another cell line such as DU145. If similar outcomes are observed, discuss the advantages and implications of using this method. Additionally, distinguish between 2D and 3D conditions in the images for clarity.

8.      Figure 5.2: It would be informative to highlight the distinctions between results obtained in 2D and 3D conditions. Does Nicotine exhibit any effects in 2D culture, and were similar trends observed in both 2D and 3D conditions?

9.      Consider including a troubleshooting section. This could provide valuable guidance for researchers encountering challenges when attempting to replicate the described protocol.

Reviewer 3 Report

Review on the manuscript
"A Simple and Fast Method for the Formation and Downstream Processing of Cancer-Cell-Derived 3D Spheroids: An Example using Nicotine-Treated A549 Lung Cancer 3D Spheres"
by
Irida Papapostolou, Florian Bochen, Christine Peinelt and Maria Constanza Maldifassi
This paper describe an original, efficient technique to generate spheroids using different cancer cells, in ultra low adhesion plates (ULA). The authors further provide protocols to test viability, and marker genes expression (qPCR). An example of nicotine exposure modification on spheroids prepared with A549 cells, as compared to spheres produced with a prostate cancer line DU145 is shown. It is showwn that low levels of nicotine (10µM) increases viability, due to the activitation of nAChR.
Overall the paper is well written and contains ineresting information, in particular the generation of spheres. Further analyses are more classical but can be done using such spheroids.
I recommend publication with the minor changes below:

MINOR CHANGES
- The authors should say how the optimal number of cells is determined. Does the size matter for good viability ?
- page 2: what does 3Rs mean ?
- page 2: (i)-(ii)-(iii) note that also mechanical properties can be measured, as a useful tool to test tumour plasticity (for example cite Tsvirkun et al. J Biomech 2022)
- page 2: effective should be efficient ?
- Figure 3E. What is the scale ?

Reviewer 4 Report

In this manuscript, the authors described their protocol for the generation of 3D spheres using ultra-low attachment (ULA) multi-well plates. Using this protocol, spheroids of different human cancer cell lines were obtained and characterized on the basis of their morphology, viability, and expression of specific markers. They described the characterization of nicotine-treated A549 lung cancer 3D spheres as an example. The specific comments are listed below.

1.      The protocol described in this manuscript is simple to reproduced in any laboratory, but the authors failed to emphasize what is the improvement in comparison to other protocols. Or what is the uniqueness in this protocol.

2.      The characterization of nicotine-treated A549 Lung Cancer 3D Spheres could be performed in any 3D culture protocol. What is the benefit of using this protocol?

3.      Did the optional centrifugation step always improve the efficiency (or speed) of sphere formation without any disadvantage? If that is the case, the authors should recommend always including this step in the protocol, since it was not a complicated procedure.

None

Round 2

Reviewer 2 Report

The author addressed all my questions. I think that this article meets the standards required by this journal.